# Isotropic Two-Dimensional Differentiation Based on Dual Dynamic Volume Holograms

Pin Wang [1,2], Houxin Fan [3], Yaping Zhang [1,*], Yongwei Yao [1], Bing Zhang [1], Wenlong Qin [1] and Ting-Chung Poon [1,4]

[1] Yunnan Provincial Key Laboratory of Modern Information Optics, Kunming University of Science and Technology, Kunming 650500, China
[2] Kunming Power Supply Bureau, Yunnan Power Grid Co., Ltd., Kunming 650011, China
[3] Center for Optical and Electromagnetic Research, State Key Laboratory for Modern Optical Instrumentation, Zhejiang University, Hangzhou 310058, China
[4] Bradley Department of Electrical and Computer Engineering, Virginia Tech, Blacksburg, VA 24061, USA
[*] Correspondence: yaping.zhang@gmail.com

**Abstract:** We study the use of two dynamic thick holograms to realize isotropic two-dimensional (2D) differentiation under Bragg diffraction. Acousto-optic modulators (AOMs) are used as dynamic volume holograms. Using a single volume hologram, we can accomplish a first-order derivative operation, corresponding to selective edge extraction of an image. Since the AOM is a 1D spatial light modulator, filtering of the image only occurs along the direction of the sound propagation. To achieve 2D image processing, two AOMs are used within a Mach–Zehnder interferometer (MZI). By aligning one AOM along the *x*-direction on the upper arm of the interferometer and another AOM along the *y*-direction on the lower arm, we accomplish the sum of two first-derivative operations, leading to isotropic edge extraction. We have performed both computer simulations and optical experiments to verify the proposed idea. The system provides additional operations in optical computing using AOMs as dynamic holograms.

**Keywords:** acousto-optic modulators; Bragg diffraction; Mach–Zehnder interferometer; isotropic edge extraction; optical computing

## 1. Introduction

The edge information in images usually carries characteristic information about objects. Indeed, edge extraction technology [1–4] has always been an important research topic in image processing. It has applications such as image enhancement [5], image restoration [6], and image segmentation [7]. Various edge extraction operators in digital image processing [8] have been used to obtain edge information. Recent optical edge extraction techniques include the use of a vortex beam in a standard 4-f coherent imaging system [9,10]. In the context of holographic image processing, Pan et al. have proposed a method for edge extraction using time-varying vortex beam in optical scanning holography for incoherent image processing [11].

Acousto-optic interaction provides a powerful means for optical information processing. Through Bragg diffraction, acousto-optic interaction can control the spatial frequency of light signals [12,13]. Compared to digital image processing techniques, using acousto-optic interaction in optical imaging systems can achieve dynamic and real-time programmable laser beam modulation [14–17]. An acousto-optic modulator (AOM) is considered as a dynamic thick hologram and is a type of spatial light modulator based on acousto-optic interaction [18], which is widely used in the real-time processing of light waves due to its high reliability, fast response speed, and programmability. The use of acousto-optic transfer functions has been used to describe the profile of the diffracted light field in acousto-optic interaction between light waves and a sound column. By multiplying the Fourier transform

of the image to be processed by the acousto-optic transfer function in the frequency domain [15] and performing an inverse Fourier transform, a processed image can be obtained. Balakshy first pointed out the use of acousto-optic interaction to control the optical image structure [19], and Xia et al. [20] were the first to perform image edge extraction using an AOM that operated under Bragg conditions in an experiment, achieving first-order differentiation operation in a given spatial direction, e.g., $\frac{\partial}{\partial x}$. The use of dual AOMs can be effectively used for various differential operations. Cao et al. [21] used two cascaded AOMs to achieve the effect of a second-order differential operation, e.g., $\frac{\partial^2}{\partial x^2}$. Banerjee et al. [22] used two orthogonally modulated cascaded AOMs to achieve mixed-direction differential operation, e.g., $\frac{\partial^2}{\partial x \partial y}$. Recently, Zhang, et al. [23] provided a detailed review of various image processing methods using AOMs, and in addition, they achieved single-sided notch filtering by deviating the incident angle from the Bragg angle. In passing, we want to point out that Voloshinov [24] first used anisotropic Bragg diffraction in crystals to perform image edge extraction.

In certain cases, it is necessary to focus on edge information in anisotropic as well as isotropic manners [25,26]. Previous research utilizing dual AOMs for edge extraction had limitations and could not perform isotropic edge extraction. A Mach–Zehnder interferometry (MZI) optical system based on two AOMs has been proposed to overcome such limitations [23] but never been implemented. This paper aims to implement isotropic and anisotropic edge extraction using two acousto-optic modulators (AOMs) in a Mach–Zehnder interferometer. For isotropic edge extraction, we implement specifically the operation of $\frac{\partial}{\partial x} + \frac{\partial}{\partial y}$ optically, which is not achievable in earlier dual-AOM systems mentioned.

## 2. Image Processing with One AOM

### 2.1. Principle of AOM and Acousto-Optic Transfer Function

Due to the acousto-optic interaction between light and sound in the AOM, the diffracted light contains the result of the modulation of the light signal. When the acoustic wave propagating in the medium is a plane wave, and the angle between the incident plane wave of light and the *z* axis satisfies Equation (1), we have Bragg diffraction. The angle of incidence $\phi_{inc} = \phi_B$ is called the Bragg angle. As shown in Figure 1, after acousto-optic interaction, only two beams of diffracted light, the 0th- and 1st-order diffraction, are produced. It can be seen that the incident light and the 0th-order beam have the same direction, and the 0th-order beam differs in direction by $2\phi_B$ from the 1st-order beam. In practice, we do not have plane wave of sounds but have a finite sound column of width *L* as shown in Figure 1. Under this situation, the sound fields spread as they propagate in the medium. Acousto-optic interaction occurs even when the direction of the incident plane wave of light is not exactly at $\phi_B$, generating multiple orders of light beam.

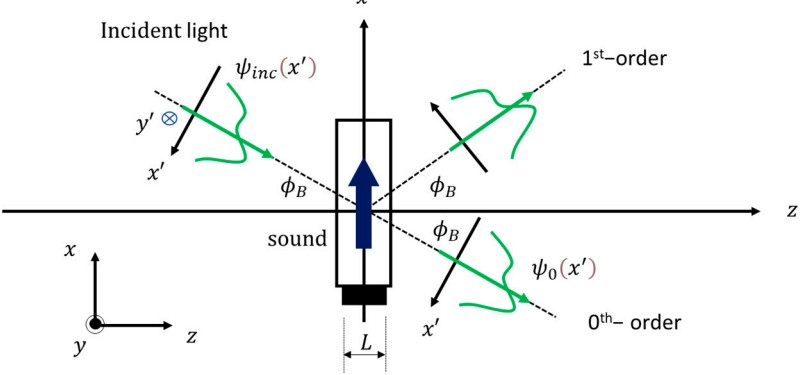

**Figure 1.** Acousto-optic modulator under Bragg diffraction. (P.S.: green arrows: direction of incident light, 1st order and 0th order light; dark blue arrow: direction of propagating sound waves).

The Bragg angle $\phi_B$ is determined by the wavelength of the acoustic wave and the incident light [12,18]:

$$\phi_B = \sin^{-1}\left(\frac{\lambda}{2\Lambda}\right),\tag{1}$$

where $\lambda$ is the wavelength of the incident light in the acoustic medium, and $\Lambda$ is the wavelength of the acoustic waves.

When an optical image is incident at the Bragg angle, we can describe the interaction of the image and the AOM by using a spatial transfer function. Figure 1 depicts the interaction between the incident light field $\psi_{inc}(x')$ and the sound waves.

The transfer function $H_0$ for the 0th-order beam can be expressed as [23]

$$H_0\left(\frac{k'_x\Lambda}{\pi}\right) = \frac{\mathcal{F}\{\psi_0(x')\}}{\mathcal{F}\{\psi_{inc}(x')\}}$$
$$= \exp\left(\frac{-jQ\Lambda k'_x}{4\pi}\right)\left\{\cos\left[\left(\frac{Q\Lambda k'_x}{4\pi}\right)^2 + \left(\frac{\alpha}{2}\right)^2\right]^{\frac{1}{2}} + \left(\frac{jQ\Lambda k'_x}{4\pi}\right)\sin c\left\{\left[\left(\frac{Q\Lambda k'_x}{4\pi}\right)^2 + \left(\frac{\alpha}{2}\right)^2\right]^{\frac{1}{2}}\right\}\right\}.\tag{2}$$

Here, $\mathcal{F}$ denotes the Fourier transform; $\mathcal{F}\{\psi_{inc}(x')\}$ represents the spectrum of the incident light entering the AOM with $k'_x$ being the spatial radian frequency along the $x'$-direction; and $\mathcal{F}\{\psi_0(x')\}$ represents the spectrum of the output 0th-order diffracted light. $\alpha$ denotes the peak phase delay of the light due to the passage of the sound wave, which is proportional to the sound pressure within the AOM. Additionally, $\sin c(x) = \frac{\sin(x)}{x}$, $j = \sqrt{-1}$, and finally, $Q = 2\pi L\frac{\lambda}{\Lambda^2}$ is the Klein–Cook parameter [27].

Equation (2) shows that image processing is along the $x'$-direction. Note that the $x$-direction and the $x'$-direction is basically the same, given a small Bragg angle typically for AOMs.

We have generated a circle pattern of $1024 \times 1024$ pixels as the original input object by assuming that the incident beam is of two transverse dimensions, i.e., $\psi_{inc}(x', y') = O(x', y')$ in the simulation. The diameter has 500 pixels corresponding to 4 mm. The wavelength of the incident light is $\lambda$ = 532 nm, and the refractive index of the acousto-optic crystal in the AOM is 2.3, giving $\phi_B \approx 3.85$ mrad. The carrier frequency of the AOM used is 120 MHz, and the wavelength of the acoustic wave in AOM is $\Lambda = 0.03$ mm. The peak phase delay of the light through the acousto-optic medium $\alpha$ is chosen to be $\pi$. Finally, the width of the piezoelectric transducer providing the sound wave signal at the lower end of the AOM is $L = 8$ mm, giving the Klein–Cook parameter $Q = 14$. Figure 2a shows the circle pattern, and Figure 2b gives the output according to Equation (3) for two transverse dimensions:

$$\psi_0(x', y') = \mathcal{F}^{-1}\left\{H_0\left(\frac{k'_x\Lambda}{\pi}\right)\cdot\mathcal{F}\{O(x', y')\}\right\}.\tag{3}$$

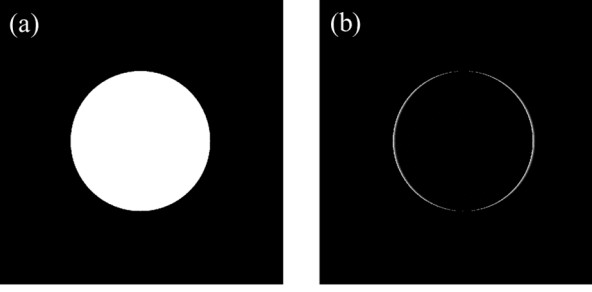

**Figure 2.** (**a**) Original input object; (**b**) Output according to Equation (3).

Clearly, we see that there is an edge extraction along the $x'$-direction. As the acoustic waves propagate along the $x$-direction, the filtering effect is along the $x'$-direction [see Figure 1], and thus, no edge information is extracted along the $y'$ direction. The edge extraction effect is unidirectional (anisotropic), and the filtering effect of a single AOM

provides approximately a first-order differential operator, which can be explained as follows.

When $\left|\left(\frac{Q\Lambda k'_x}{4\pi}\right)\right| \ll \left|\frac{\alpha}{2}\right|$, Equation (2) becomes

$$H_0\left(\frac{k'_x\Lambda}{\pi}\right) = \exp\left(\frac{-jQ\Lambda k'_x}{4\pi}\right)\left\{\cos\left(\frac{\alpha}{2}\right) + \left(\frac{jQ\Lambda k'_x}{4\pi}\right)\sin c\left(\frac{\alpha}{2}\right)\right\}. \tag{4}$$

Now, with $\cos\left(\frac{\alpha}{2}\right) = 0$ condition, the above equation becomes

$$H_0\left(\frac{k'_x\Lambda}{\pi}\right) = \exp\left(\frac{-jQ\Lambda k'_x}{4\pi}\right)\left\{\left(\frac{jQ\Lambda k'_x}{4\pi}\right)\sin c\left(\frac{\alpha}{2}\right)\right\} = \exp\left(\frac{-jQ\Lambda k'_x}{4\pi}\right)jBk'_x, \tag{5}$$

where $B = \left(\frac{Q\Lambda}{4\pi}\right)\sin c\left(\frac{\alpha}{2}\right)$ is a constant. Finally, we arrive at Equation (6)

$$H_0\left(\frac{k'_x\Lambda}{\pi}\right) \propto \exp\left(\frac{-jQ\Lambda k'_x}{4\pi}\right)(jk'_x) \approx jk'_x. \tag{6}$$

From Equation (6), it can be seen that the 0th-order diffracted light is the result of high-pass filtering of the input light profile. Since $\mathcal{F}\left\{\frac{\partial\psi_{inc}(x')}{\partial x'}\right\} = -jk'_x\mathcal{F}\{\psi_{inc}(x')\}$, this indicates that the transfer function $H_0\left(\frac{k'_x\Lambda}{\pi}\right)$ in Equation (3) will have the effect of a first-order differential operation, which is evidenced by combining Equations (3) and (6) to obtain, with $\psi_{inc}(x') = O(x', y')$,

$$\psi_0(x', y') = \mathcal{F}^{-1}\left\{H_0\left(\frac{k'_x\Lambda}{\pi}\right)\cdot\mathcal{F}\{\psi_{inc}(x', y')\}\right\} = \mathcal{F}^{-1}\{jk'_x\mathcal{F}\{O(x', y')\}\} = \frac{\partial O(x', y')}{\partial x'}. \tag{7}$$

It should be noted that the linearity of $k'_x$ in the exponential term in Equation (6) causes a shift in the position of the processed output image, which is ignored as it is inconsequential to image processing. That is the reason why we give the final result of $jk'_x$ as the expression for $H_0\left(\frac{k'_x\Lambda}{\pi}\right)$ shown in Equation (6).

Figure 3 shows a practical version of a single-AOM optical setup. The incident light is emitted from the laser, passes through the pinhole, and is transformed into a plane wave by the collimating lens L1. It then passes through the object $O(x', y')$, which serves as the system input in the form of the incident light beam profile, and the AOM is placed between the object plane and imaging lens L2, which images the object onto the CCD. An Iris is used in the system to select the different diffraction orders. We shall use Figure 3 as part of the system to implement anisotropic filtering in the subsequent section.

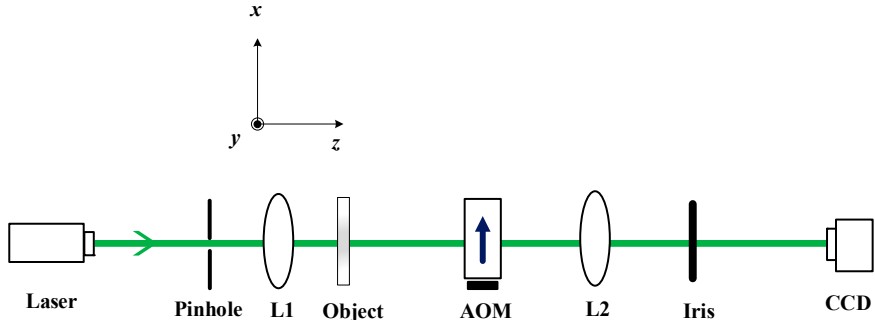

**Figure 3.** Practical single-AOM optical setup. (P.S.: green arrow: direction of incident light; dark blue arrow: direction of propagating sound waves).

## 2.2. Rotation of AOM on the $x'-y'$ Plane

To demonstrate the anisotropic edge extraction capability of the acousto-optic modulation system, we rotate the AOM around the $z$-axis by an angle $\theta$ on the $x-y$ plane, as shown in Figure 4.

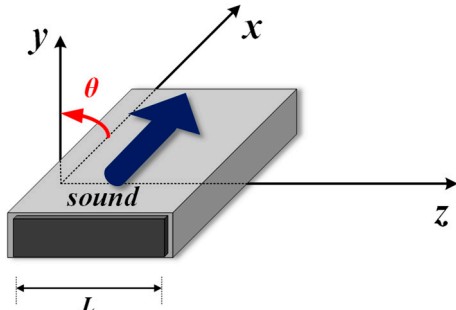

**Figure 4.** Rotation of AOM.

According to rotation transformation, the $x'$–$y'$ coordinates, accordingly, will be transformed to a new set of coordinates $x''$–$y''$ as

$$\begin{bmatrix} x'' \\ y'' \end{bmatrix} = \begin{bmatrix} \cos\theta & \sin\theta \\ -\sin\theta & \cos\theta \end{bmatrix} \begin{bmatrix} x' \\ y' \end{bmatrix}, \tag{8}$$

where $x''$ and $y''$ are the rotated coordinates. Correspondingly, the frequency domain rotation relationship is given by

$$\begin{bmatrix} k_x'' \\ k_y'' \end{bmatrix} = \begin{bmatrix} \cos\theta & \sin\theta \\ -\sin\theta & \cos\theta \end{bmatrix} \begin{bmatrix} k_x' \\ k_y' \end{bmatrix}. \tag{9}$$

So the processing direction is actually along the $x''$-direction instead of the $x'$-direction. Therefore, the transfer function $H_0\left(\frac{k_x'\Lambda}{\pi}\right)$ of the AOM modulation on the incident light after rotation by an angle $\theta$ is now given by $H_0\left(\frac{k_x''\Lambda}{\pi}, \theta\right)$. In terms of $k_x'$, we have, using Equation (9),

$$H_0\left(\frac{k_x''\Lambda}{\pi}, \theta\right) = \exp\left(\frac{-jQ\Lambda\left(k_x'\cos\theta + k_y'\sin\theta\right)}{4\pi}\right) \cdot \left\{ \cos\left[\left(\frac{Q\Lambda\left(k_x'\cos\theta + k_y'\sin\theta\right)}{4\pi}\right)^2 + \left(\frac{\alpha}{2}\right)^2\right]^{\frac{1}{2}} + \left(\frac{jQ\Lambda\left(k_x'\cos\theta + k_y'\sin\theta\right)}{4\pi}\right)\sin c\left\{\left[\left(\frac{Q\Lambda\left(k_x'\cos\theta + k_y'\sin\theta\right)}{4\pi}\right)^2 + \left(\frac{\alpha}{2}\right)^2\right]^{\frac{1}{2}}\right\}\right\}, \tag{10}$$

and the processed image is given by

$$\psi_0(x', y') = \mathcal{F}^{-1}\left\{H_0\left(\frac{k_x''\Lambda}{\pi}; \theta\right)\mathcal{F}\{O(x', y')\}\right\}. \tag{11}$$

Figure 5a,b show the output when the AOM is rotated for $\theta = \pi/4$ and $\theta = -\pi/4$, respectively. By adjusting the angle of the AOM rotation, anisotropic edge extraction can be achieved according to the angle of rotation.

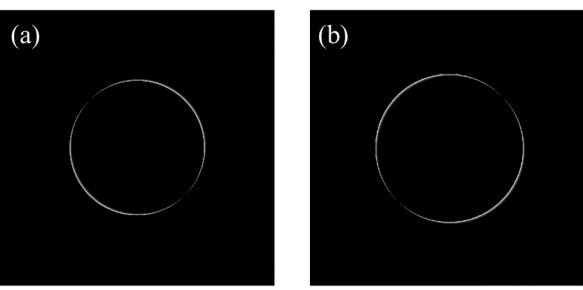

**Figure 5.** (**a**) $\theta = \pi/4$; (**b**) $\theta = -\pi/4$.

## 3. Mach–Zehnder Interferometer (MZI) with Dual Acousto-Optic Modulators

From the simulations in the last section, it can be seen that when using a single acousto-optic modulator (AOM) to modulate the image, the edge information of the incident light $O(x', y')$ approximately along the direction of the acoustic wave in the AOM is extracted, and there is no filtering effect in the direction that is perpendicular to the direction of sound, giving anisotropic filtering. In the field of image processing, in some cases, we hope to obtain isotropic filtering. To solve this problem, two orthogonally oriented AOMs within the two arms of a Mach–Zehnder interferometer have been proposed [23]. The system is shown in Figure 6.

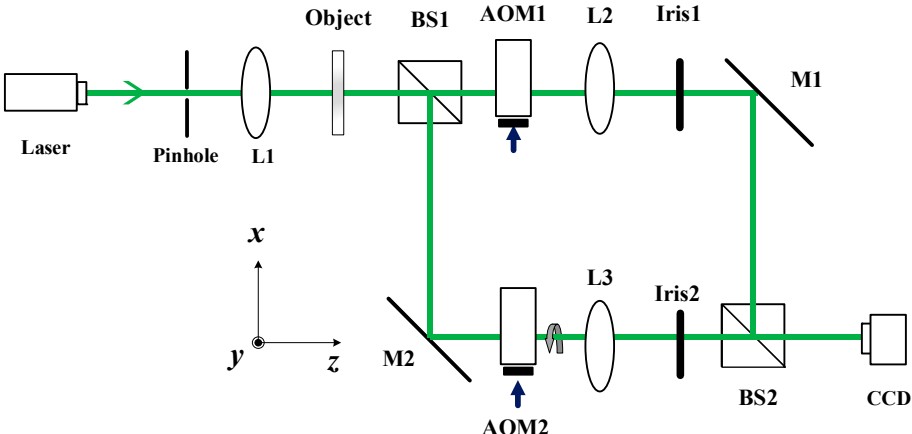

**Figure 6.** Dual AOM optical system using a Mach–Zehnder interferometer. (P.S.: green arrow: direction of incident light; dark blue arrow: direction of propagating sound waves).

Within the interferometer, each arm uses an AOM for modulation. The incident light is emitted from the laser, passes through the pinhole and is transformed into a plane wave by the collimating lens L1. The plane wave then illuminates the input object $O(x', y')$, which is then split by a 1:1 beam splitter (BS1). The two split beams are modulated by AOM1 and AOM2, respectively, under the Bragg condition. L2 and L3 are imaging lenses that image the input object onto the CCD. Iris1 and Iris2 are used to select the zeroth-order beams. The beamsplitter BS2 is used to add the two images from the two arms.

In principle, the AOM can be rotated arbitrarily on the $x$–$y$ plane as shown in Figure 1. Assuming that the travelling sound waves of AOM1 in the upper arm of the interferometer are along the $x$-direction ($\theta = 0$), and the sound waves of AOM2 in the lower arm are along the $y$-direction ($\theta = \pi/2$), we can obtain the filtered light field

$$
\begin{aligned}
\psi_0(x', y') = \mathcal{F}^{-1}\left\{ H_0\left(\frac{k_x'' \Lambda}{\pi}; \theta = 0\right) \mathcal{F}\{O(x', y')\} \right\} \\
+ \mathcal{F}^{-1}\left\{ H_0\left(\frac{k_x'' \Lambda}{\pi}; \theta = \frac{\pi}{2}\right) \mathcal{F}\{O(x', y')\} \right\}.
\end{aligned}
\tag{12}
$$

Figure 7a,c show the original input for a circular and a rectangular input profiles. Figure 7b,d show the corresponding outputs. The same AOM parameters have been used as in the simulations for Figure 2.

Some optical experiments have also been performed using the dual-AOM MZI system shown in Figure 6. The laser is a green light with wavelength of 532 nm. The pinhole and collimating lens L1 are used to collimate light into plane waves. The circular pattern on the object plane has a diameter of 4 mm. The refractive index of the AOM crystal is 2.3, and the carrier frequency of the AOM is 120 MHz. The focal length of imaging lens L2 and L3 is 150 mm. The photographs (by MMRY UC900C CCD camera) of the experimental results are shown in Figure 8. Figure 8a shows the anisotropic edge extraction result of the circular pattern with only AOM1 working at a rotation angle of $\theta = -\pi/4$ based on Equation (11), in which it can be observed that the edge information along the $\theta = \pi/4$

direction was not preserved, consistent with the simulation result in Figure 5b. Figure 8b shows the isotropic edge extraction result of the rectangular pattern using the dual AOMs in the *x*- and *y*-directions. The rectangular pattern has dimensions of 4 mm × 4 mm.

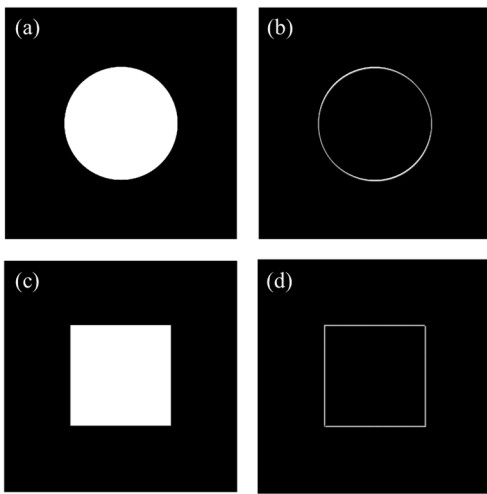

**Figure 7.** Using two AOMs with one in the *x*-direction and the other in the *y*-direction: (**a**,**c**) original input; (**b**,**d**) output beam profiles corresponding to (**a**,**c**).

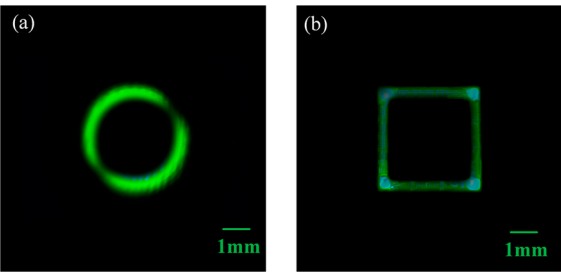

**Figure 8.** Results of optical experiment: (**a**) output profile for an input circular pattern; (**b**) output profile for the rectangular beam profile.

We observe that in the output in Figure 8b for the rectangular pattern, there are edge extractions along the *x*- and *y*-directions, but in addition, there seem to be bright spots on the four corners, which is not fully consistent with the result shown in Figure 7d. However, this discrepancy can be fully explained if we rewrite Equation (12) and employ the result of Equation (7) for each direction to obtain

$$\psi_0(x', y') = \frac{\partial O(x', y')}{\partial x'} + \frac{\partial O(x', y')}{\partial y'}, \tag{13}$$

and this clearly supports the result shown in Figure 7d. Now, since the CCD only captures intensity, i.e., $|\psi_0(x', y')|^2$, what is displayed on Figure 8b is then

$$|\psi_0(x', y')|^2 = \left| \frac{\partial O(x', y')}{\partial x'} + \frac{\partial O(x', y')}{\partial y'} \right|^2$$
$$= \left| \frac{\partial O(x', y')}{\partial x'} \right|^2 + \left| \frac{\partial O(x', y')}{\partial y'} \right|^2 + 2 \frac{\partial O(x', y')}{\partial x'} \frac{\partial O(x', y')}{\partial y'}. \tag{14}$$

Note that the first two terms give the results shown in Figure 9b. However, the fixed derivative will extract the four corners and therefore give bright spots at the corners. The superposition of all the terms in Equation (14) gives the result shown in Figure 9c in simulations and in Figure 8b from the optical experiment, where the four corners of the

square are also emphasized. We, therefore, have consistent results both from simulations as well as optical experiments.

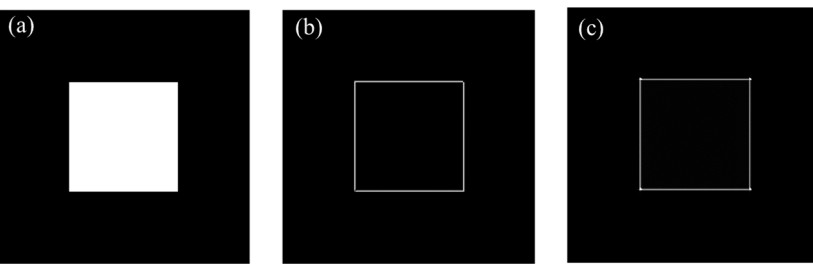

**Figure 9.** Results of simulations of a rectangular object. (**a**) Original rectangular input object, (**b**) output field distribution $\psi_0(x', y')$ according to Equation (12), (**c**) output intensity $|\psi_0(x', y')|^2$.

## 4. Conclusions

We have presented an AOM-based method for real-time edge extraction that enables edge extraction both isotropically and anisotropically. By rotating the AOM, anisotropic edge extraction can be achieved to extract edge information along the direction of sound propagation. We have also implemented a previously proposed Mach–Zehnder interferometric optical system based on dual AOMs, where the use of two AOMs results in a summation operation of two first-order differentiation operations previously not achievable with dual-AOM systems. We have verified our approach through both computer simulations and optical experiments.

**Author Contributions:** Conceptualization, P.W. and H.F.; methodology, software, P.W. and H.F.; validation, P.W. and W.Q.; analysis, review and editing, Y.Z., Y.Y. and B.Z.; original draft preparation, P.W. and H.F.; review and editing, T.-C.P. All authors have read and agreed to the published version of the manuscript.

**Funding:** The authors would like to acknowledge the support of this work by the National Natural Science Foundation of China (Grant No. 62275113), Yunnan Provincial Science and Technology Department (Xing Dian Talent Support Program), Research Project of Research Center for Analysis and Measurement Kunming University of Science and Technology (Grant No. 2021P20193103002), and Youth Fund of Yunnan Provincial Department of Science and Technology (Grant No. 202201AU070159) from Yunnan Province.

**Data Availability Statement:** The data that support the results within this paper and other findings of the study are available from the corresponding authors upon reasonable request.

**Conflicts of Interest:** The authors declare no conflict of interest.

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
