# Peer review of "Isotropic Two-Dimensional Differentiation Based on Dual Dynamic Volume Holograms"

_photonics, doi:10.3390/photonics10070828_

Round 1

Reviewer 1 Report

Analytical expressions that are the basis of work (2) and (4) are not sufficiently well-grounded. A detailed explanation of the transition from (2) to (4) is needed. In my time, I have worked with acousto-optic light modulators under Bragg conditions, but I have not seen anything like this. In the paper, it is necessary to specify how to calculate kx (ky) and within what limits it can vary.

Reviewer 2 Report

Your paper entitled, Isotropic Two-Dimensional Differentiation Based on Dual Dynamic Volume Holograms, is well written. The image processing community can benefit from an isotropic, two-dimensional differentiation technique that enables real-time edge extraction both isotopically and anisotropically. Your novel method, which implements isotropic and anisotropic edge extraction using two acousto-optic modulators (AOMs) in a Mach-Zehnder interferometer (MZI) to achieve 2D image processing, is an advancement over current methods. A MZI optical system based on two AOMs has been previously proposed to perform isotropic edge extraction, but never implemented. You implement two dynamic thick holograms to realize isotropic two-dimensional (2D) differentiation under Bragg diffraction.

I made a couple of punctuation suggestions (See the attached PDF document – Comments / Recommendations are in Sticky Notes).

Quality of English is good.

Author Response

Thank you so much for your careful check. We have revised according to your sticky notes.

Reviewer 3 Report

The manuscript entitled ” Isotropic Two-Dimensional Differentiation Based on Dual 2 Dynamic Volume Holograms    by   Yaping Zhang  et al. reports the use of two dynamic thick holograms to realize isotropic two-dimensional (2D) differentiation under Bragg diffraction and Acousto-optic modulators (AOMs) are used as dynamic volume holograms. This research work might continue the author's article "Optical image processing using acousto-optic modulators as programmable volume holograms: a review [Invited]". The research work is awe-inspiring, and this study of the edge extraction method is much needed in the holographic regime. The authors have performed quite good research on this investigation. However, the article needs some clarification. The language used in this article is reasonable. The critical statement to support the results/finding, and the citation for the statement are missing. Overall, the article is worthy. However, all the results are needed to connect, and a thorough description by providing detailed information can improve the fineness of this manuscript. I want to address a few queries on this manuscript, which will help improve the quality of the article. Please find the comment below.

1.      What is the strong motivation for this research work? And it needs to  include in the introductory section

2.      Page no. 2, line 84, “The Bragg angle ?? is determined by the wavelength of the acoustic wave and the 84 incident light: [*]. Please provide any suitable references if you have any.

3.      Figure 2 (a), what does the author mean by the original object? In the description, it is mentioned as a circle pattern.

  1. Page no. 4, line 118, What is the physical significance of Klein_Cook Parameter? Why did the author take the value of Q=14? 
  2. How does the author rotate the AOM since the diffraction angle is so sensitive and the AOM is connected to the driver? 
  3. What is the diffracted beam's intensity variation after the AOM's rotation? 
  4. In Figure 6, what is the modulation difference in the arms of the Dual AOM Mach-Zender interferometer? 
  5. Figure 6, How the experimental set-up differs from the previously reported article by the author [23]?
  6. What software is used for the simulation?

Reviewer 4 Report

This paper reports on  two-dimensional edge extraction of optical images with the help of   acusto-optical modulators placed in the two arms of a Mach-Zehnder interferometer. The theoretical part is based on the authors' earlier review  (Chin. Opt. Lett. 2022,20, 29-38); the experimental demonstration is, however, original. The paper is well written, the optical experiments are in reasonable agreement with the simulations. 

A remark: At line 111 it is stated that    " the incident beam is of two transverse dimensions, i.e., ????(?′,?′)=?(?′,?′)". It should be explained in more details, what does these mean, and how is the  ?(?′,?′) function defined. This would be important, so much the more as the function is often cited in the rest of the paper.

Round 2

Reviewer 3 Report

The authors have submitted a revised manuscript entitled ”Isotropic Two-Dimensional Differentiation Based on Dual  Dynamic Volume Holograms" by Yaping Zhang et al. reports the use of two dynamic thick holograms to realize isotropic two-dimensional (2D) differentiation under Bragg diffraction and Acousto-optic modulators (AOMs) are used as dynamic volume holograms. The authors have significantly improved the revised manuscript by illustrating their research regarding modifications of contents and equations, and texts. Also, the authors have given satisfactory responses to the comment raised. The article will be the appropriate form for publication.